# An Innovative Small-Target Detection Approach Against Information Attenuation: Fusing Enhanced Programmable Gradient Information and a Novel Mamba Module

**DOI:** 10.3390/s25072117

**Published:** 2025-03-27

**Authors:** Yang Liu, Yatu Ji, Qingdaoerji Ren, Bao Shi, Na Liu, Min Lu, Nier Wu

**Affiliations:** 1School of Information Engineering, Inner Mongolia University of Technology, Hohhot 010051, China; 20221100134@imut.edu.cn (Y.L.); csnaliu@imut.edu.cn (N.L.); cslumin@imut.edu.cn (M.L.);; 2Department of Mechanical and Electrical Engineering, Inner Mongolia Vocational College of Chemical Engineering, Hohhot 010080, China

**Keywords:** small object detection, vision mamba, programmable gradient information

## Abstract

Serious information loss often occurs when the input data undergoes the layer-by-layer feature extraction process through deep neural networks. In small-target detection tasks, this problem becomes more serious. The existing theory holds that the information bottleneck leads to the decrease of an algorithm’s recognition rate. In order to improve the precision of small-target recognition, PGI-ViMamba is proposed in this paper to resist the information attenuation of a neural network model. The model mainly uses Multi-Level Attention-Gated Programmable Gradient Information (MLAG PGI) and SPD-Conv-VSS module. Specifically, the backbone uses an improved VSS module as a feature extractor and uses an auxiliary branch similar to YOLOv9. This neural network structure ensures that the neural network retains the feature information of the small target in the process of forward propagation while reducing the model parameters. Another advantage is that the MLAG PGI acts as a reversible branch, providing reliable gradient information. Experiments show the effectiveness of the algorithm. Compared with state-of-the-art (SOTA) models, the proposed method achieves improvements of 1.1% and 2.1% in small-target recognition precision on the VisDrone and DOTA-v1.5 datasets, respectively, with no significant decline in recall rates. Additionally, ablation experiments validate the effectiveness of the MLAG PGI and SPD-Conv-VSS modules.

## 1. Introduction

Small-target recognition represents a critical challenge within the realm of computer vision, with the objective of identifying objects that are located at a significant distance from the observer within a vast scene. The inherent characteristics of this task, such as the subtlety of the features of the objects to be identified and the presence of an information bottleneck in existing models, contribute to the increased difficulty of detecting small targets. In domains such as aerial target detection, unmanned aerial vehicle (UAV) applications, and industrial inspection, there is an urgent need to enhance the neural network’s capacity for recognizing small targets. In the field of deep learning, the detection of small objects presents a formidable challenge due to the limited pixel representation of these objects within an image, which can impede the model’s ability to effectively capture the requisite features for precise identification. Nevertheless, researchers have devised a variety of strategies to surmount this obstacle. Among the commonly employed techniques are the following:1.Image Pyramids and Feature Pyramids [1,2].

An image pyramid generates a series of images of different resolutions by downsampling or upsampling the original image. The images are arranged in order of resolution from highest to lowest, forming a pyramid-like structure. A feature pyramid generates a series of feature maps of different scales through multi-scale processing of images or feature maps. These feature maps are arranged on a scale from large to small, forming a pyramidal structure. The above two techniques are widely used in small-target recognition. Both capture different information about an image or feature through multi-scale representation. An image pyramid realizes multi-scale representation at the pixel level, which is conducive to preserving the fine-grained information of the image. A feature pyramid realizes multi-scale representation at the feature level and is good at extracting the semantic information of an image.

2.Oversampling [3,4,5].

Oversampling is a technique that increases the proportions of small objects in the image by repeatedly sampling regions of the image containing small objects. This can enhance the representation of small objects in the feature map, thereby improving detection accuracy.

3.Anchor boxes [6,7,8].

ROI Pooling plays a crucial role in small object detection by converting regions of interest (ROIs) of varying sizes into fixed-size feature maps, enabling subsequent classification or regression tasks. This method was first introduced in Faster R-CNN [8] and has since been widely applied in the field of object detection. The core function of ROI Pooling in small object detection is to map small-sized ROIs to a fixed-size output, ensuring that regardless of the original size of the ROI, a uniform-sized feature representation is obtained. This allows subsequent fully connected layers to accept these features and perform effective classification or regression operations.

Recently, the introduction of Vision Mamba (ViM) [9] and Programmable Gradient Information (PGI) [10] has provided new possibilities for improving the accuracy of small object detection. These two technologies have driven the development of object detection algorithms on different levels.

The main motive of this paper is to reduce the influence of the information bottleneck effect on feature extractors and improve the precision of recognition of small targets. Our main task is to propose the PGI-ViMamba model, which is an innovative fusion of the visual state space (VSS) block [11] and PGI techniques for object detection. The PGI-ViMamba model aims to enhance the accuracy and efficiency of detecting small objects within images.

The main contributions of this paper are as follows:

1. The present work achieves the integration of Vision Mamba and PGI, effectively leveraging their respective strengths in the task of small-target recognition. This results in a comprehensive solution for detecting small objects in images.

2. We have developed the SPD-Conv-VSS module, which enhances the feature extraction capability of small-scale targets by modifying the VSS module of the ViM model. This enhancement facilitates the network in capturing comprehensive and intricate feature information, thereby playing a crucial role in detecting small objects.

3. We specially designed the Multi-Level Attention-Gated PGI to provide reliable gradient information for the backbone. The improvement of this structure effectively solves the problem of information bottlenecking in deep neural networks and ensures a more accurate correlation between the detected object and the input during training. This advance is particularly valuable for algorithms that detect tiny targets.

## 2. Related Work

### 2.1. Mamba and Vision Mamba

Mamba [11] is an advanced state space model (SSM) [12] designed for efficient processing of complex data-intensive sequences, and its initial successful application was in the field of NLP. One of the main advantages of Mamba is its ability to solve the computational challenges associated with handling long sequences of conventional transformers. The selective approach of Mamba significantly speeds up inference, has a throughput five times higher than a standard transformer, and shows linear complexity with sequence length in terms of computation. Notably, even when the sequence extends to one million elements, the performance of Mamba continues to improve with increasing actual data.

Inspired by Mamba and ViT [13], ViM is a deep learning model specifically designed for computer vision tasks to improve the efficiency and effectiveness of visual representation learning. It aims to improve computational efficiency while maintaining a global reception field. The innovation of the ViM model lies in its ability to efficiently convert two-dimensional images into a sequence form suitable for processing and capture spatial information through bidirectional sequence modeling. This model addresses the issue of direction sensitivity by introducing the 2D selective scanning module (SS2D) [9], which converts images into ordered patch sequences, achieving linear complexity without sacrificing the global reception field. The ability of ViM to capture rich and detailed feature information is particularly important for small object detection. Specifically, the ViM model first divides a two-dimensional image into a series of small patches, then flattens these patches into a one-dimensional sequence and performs linear projection to obtain a fixed-size vector representation. During this process, ViM also adds positional embedding information to preserve the spatial layout information of the image. The core component of the ViM model is the VSS block, which is specifically designed for vision tasks and integrates the ability of bidirectional sequence modeling. Compared to the original Mamba model, the ViM block is more suitable for processing vision tasks that require spatial awareness. The operation algorithm of the VSS block takes into account the spatial dimension of the image, making it more efficient and accurate in processing visual data. In terms of architectural details, the ViM model contains multiple hyperparameters, such as the number of blocks, which can be adjusted according to specific vision tasks to achieve optimal performance. Previous studies have shown that ViM models provide powerful tools for computer vision tasks through their efficient visual representation learning and processing capabilities. At present, it has been successfully applied in many visual tasks.

### 2.2. YOLOv9 and Programmable Gradient Information (PGI)

YOLOv9 [10] is an important real-time object detection model following the YOLO series [14,15,16,17,18,19] and represents the latest progress of the YOLO series. YOLOv9 incorporates several innovations, including the concept of Programmable Gradient Information (PGI). PGI consists of three parts: the backbone branch, auxiliary reversible branch, and multi-level auxiliary information. The main branch is usually a PANET [20]-like structure. The core of PGI is to reduce losses in the information layer-by-layer modeling by introducing auxiliary gradient branches to calculate losses and gradients. The auxiliary gradient branches only participate in the calculation of losses in training. The auxiliary reversible branch provides the supervision mechanism and gradient information to ensure the correctness and effectiveness of the main branch information modeling. Multi-level auxiliary information enables all three layers of features to receive all of the information of large and small targets, avoiding the loss of target information in the deep feature pyramid.

By providing complete input information for the target task to compute the objective function, PGI can generate more reliable gradient information to update network weights, thereby helping the model learn better.

## 3. Method and Principle

### 3.1. Architecture Overview

The proposed PGI-ViMamba is shown in Figure 1. The design idea of the neural network is to avoid using the bottleneck structure that loses the feature information of small targets and to use SPD-Conv-VSS as the main feature extractor. PGI-ViMamba has two branches, comprising a main branch and an auxiliary branch. The auxiliary branch is mainly composed of the SPD-Conv-VSS module with four auxiliary detection heads. The main branch is mainly composed of the SPD-Conv-VSS block and a patch merging block, and it has a main detection head. Meanwhile, a multi-level gated PGI is specially designed between the main branch and the auxiliary branch, whose main function is to achieve the reverse propagation of gradient information.

In the training process, the main network and the auxiliary branch are trained together, and the auxiliary branch provides reliable gradient information for the main network. In the inference stage, the auxiliary branches is retained, and the inference process is jointly completed by the main network and the auxiliary branches.

### 3.2. SPD-Conv-VSS Module

In visual tasks, VSS blocks are often embedded in networks. As illustrated in Figure 2, to enhance the network’s capability to recognize small targets, we have improved the VSS block. It consists of two main branches. The first branch is mainly composed of the SPD-Conv [21] module, and the second branch is composed of the VSS module.

As a universal and unified module, SPD-Conv can improve the learning efficiency of feature representation without losing too much fine-grained information. The basic principle of SPD-Conv is to convert the spatial information of images into depth information, so that convolutional neural networks (CNNS) can learn image features more effectively. Specifically, the core of SPD-Conv is the Space-to-Depth operation (SDP), which compacts the original spatial dimension while remapping the values of multiple pixels to the depth dimension (i.e., the channel dimension). For example, the dimension of the feature graph is H × W × *C*, which is divided into several smaller regions. The spatial dimension of each region is reduced to H/2 × W/2, and the number of channels is increased to 4 × *C*. The principle of SPD-Conv is as shown in the following formula.(1)$$f_{i}=X[0:W:scale,0:H:scale,:],i=0,\ldots ,scale$$(2)$$output=stack\left(f_{0},f_{1},\ldots ,f_{scale}\right)$$
where X represents the feature map and stack () indicates stack operations along the channel dimension. The term scale indicates the interval at which data is fetched from the feature map. The specific process is to split the feature map by pixel intervals scale and stack the results of these splits by channel direction. With this approach, SPD-Conv is able to retain richer information in the feature extraction stage, thereby improving the model’s recognition performance for small objects and low-resolution images.

The second branch consists of the VSS block, which mainly includes SS2D. The SS2D module consists of three main parts: the scan expanding operation, the Selective Scan Space State Sequential (S6) block operation, and the scan merging operation.

As shown on the left side of Figure 2, the scan expanding module in the diagram represents the scan expanding operation. This operation flattens the input 2D feature map into a 1D vector in four distinct directions: top-left to bottom-right, bottom-left to top-right, bottom-right to top-left, and top-right to bottom-left. The operation ensures that each element in the feature map integrates information from all other positions across different directions, thereby forming a global reception field. The output of the scan expanding block consists of image patch sequences scanned along the four directions, and these four sequences are fed into four separate S6 blocks as their respective inputs. The Scan merging module in Figure 2 is responsible for reshaping and merging the four sequences processed by S6, then reconstructing them into a two-dimensional feature map as the output. Now, let us describe the principle of S6 in detail.

The S6 block in the figure is that the four one-dimensional vector sequences obtained from the previous step are independently fed into the S6 block for feature extraction. The S6 block effectively reduces quadratic complexity to linearity by selectively scanning the spatial state sequence model, interacting each element with information that has been scanned before. This step ensures that information in all directions is thoroughly scanned to capture different features.

State Space Models (SSMs) [22,23] are typically used to describe time-invariant systems. The SS2D module models image processing tasks as a discretized linear time-invariant (LTI) system operating on patch sequences. A continuous linear time-invariant system can be represented by a linear ordinary differential equation (ODE):(3)$$h^{\prime }\left(t\right)=Ah\left(t\right)+Bu(t)$$(4)$$y\left(t\right)=Ch\left(t\right)+Du(t)$$
where u(t)∈R is the map of the input signal, y(t)∈R is the output response,  h(t) is the hidden state, and $h^{\prime }\left(t\right)$ represents the derivative of h(t). $A\in R^{N\times N},B\in R^{N\times 1},C\in R^{1\times N},D\in R^{1}$ are the weighting parameters.

For deep learning-based image processing tasks, the required state transitions are discrete rather than continuous. We therefore introduce the state discretization methodology. Let the image patch sequence received by the S6 block from the scan expanding module be denoted as x. The t-th image patch can be represented as $x_{t}$. The method of discretizing the state space model using a deep neural network is as follows:(5)$$B=Linear\left(x\right),C=Linear\left(x\right)$$(6)∆ =Linear(Linear(x))

The above formulas indicate that the image patch sequence x is projected into the parameters $B\in R^{N\times 1}$, $C\in R^{1\times N}$, and $∆\in R^{N\times N}\;$ through some linear neural network Linear(). Since the linear neural network incorporates learnable parameters, the matrices B, C, and ∆ in the equations above can be regarded as learnable matrices. The following equations describe the computational process for discretizing the state space model.(7)$$\bar{A}=e^{∆\;A}$$(8)$$\bar{B}=(\bar{A}-I)A^{-1}B\approx ∆\;B$$(9)$$\bar{C}=C,\bar{D}=D$$

The equations above achieve the discretization of the matrices *A*, *B*, *C*, and *D*. The specific calculation process of the S6 module can be expressed by the following formulas:(10)$$h_{t}=\bar{A}h_{t-1}+\bar{B}x_{t}$$(11)$$y_{t}=\bar{C}h_{t}+\bar{D}x_{t}$$(12)$$y=[y_{1},y_{2},\ldots ,y_{n}]$$

Equation (10) represents the discretized state transition equation, corresponding to the continuous state transition equation in Equation (3), where $h_{t-1}$ denotes the hidden state at step *t* − 1-th and $h_{t}$ represents the current hidden state. Here, $x_{t}$ indicates the t-th image patch in the sequence x. Equation (11) corresponds to the continuous output equation in Equation (4), where $y_{t}$ is the model’s prediction for the *t*-th image patch. Finally, *y* in Equation (12) denotes the aggregated output formed by concatenating all of the predictions to integrate information across all of the image patches.

The scan merging module in the figure indicates that the scan merging operation will add and merge the sequences from all four directions, restoring the output image to the same size as the input. This step completes the integration of the global information and outputs the 2D feature map containing rich feature information.

The two branches are fused by a concat operation and dimensionally transformed by a 1 × 1 convolution at the output. The SS2D module exerts the most significant influence on the parameters in the VSS block. It serves as the core technology in ViM for achieving linear scanning of image blocks and extracting image features. The SPD-Conv-VSS module integrates SS2D and SPD-Conv, serving as a feature extraction technique highly sensitive to small targets.

### 3.3. Multi-Level Attention-Gated PGI (MLAG PGI)

As shown in Figure 3, we compare the PGI with the proposed MLAG PGI. The PGI proposed by YOLOv9 mainly consists of three parts: the main branch, the auxiliary reversible branch, and the multi-level auxiliary information. The main branch is the main part of neural network reasoning. The secondary reversible branch is used to generate reliable gradient information and provide a reverse transmission path for the primary branch. A multi-level auxiliary information control backbone is used to learn programmable multi-level semantic information. The MLAG PGI proposed in this paper makes three improvements on the basis of PGI. The first improvement is to reduce the number of detection heads on the main branch. The second improvement is to increase the number of detection heads on the auxiliary reversible branch. An important improvement is the addition of a multi-level attention-gated mechanism in the main branch and the auxiliary reversible branch, the detailed design of which can be seen in Figure 4.

Inspired by PGI, we designed an auxiliary network in parallel with the backbone network. The auxiliary network is connected to the backbone network through a backward step-down connection, which provides reliable gradient information for the main backbone network. The original PGI was modified to remove multiple levels of auxiliary information. The multi-scale feature fusion is removed from the backbone network. We believe that this improvement is conducive to improving the accuracy of small-target recognition. The principle of Multi-Level-Attention-Gated Programmable Gradient Information (MLAG PGI) can be seen in the following formulas.(13)$$V_{i}=W_{V}F_{i}$$(14)$$K_{i}=W_{K}F_{i}$$(15)$$Q_{i}=W_{Q}F_{i}$$(16)$$F_{i}^{\prime }=AGLayer\left(F_{i-1}^{\prime },F_{i}\right)=W_{\tau }\cdot \sigma \left(softmax\left(\frac{Q_{i}K_{i}^{T}}{\sqrt{d}}\right)V_{i}\right)+F_{i-1}^{\prime },i\geq 1$$(17)$$F_{i\;}=MLAG\left(x,F_{1},\ldots ,F_{i-1}\right)\;\;=\left\{\begin{array}{c} AG\;Layer(F_{0}^{\prime },F_{1}),i=1\\ AG\;Layer\left(F_{i-1}^{\prime },F_{i}\right)=AG\;Layer\left(AG\;Layer\left(F_{i-2}^{\prime },F_{i-1}\right),F_{i}\right),i\geq 2 \end{array}\right.$$

Formulas (13)–(15) are the calculation formula of $V_{i}$, $K_{i}$, and $Q_{i}$ in the attention mechanism. Formula (16) is the calculation formula of the AG Layer; its inputs are $F_{i-1}^{\prime }$ and $F_{i}$. $F_{i-1}^{\prime }$ is the output of the CBLinear module or the output of the superior AG Layer, where σ() represents the sigmoid() function and d represents the feature dimension. $F_{i}$ is the output of the MLAG of the previous level.${\;F}_{i}$ indicates the feature map of the MLAG PGI output of the previous level. It should be noted that the calculation of $F_{i}$ uses recursive calls, as seen by Equation (17). ${\;W}_{V},{\;W}_{K},{\;W}_{Q},{\;W}_{\tau }$ represent four learnable matrixes.

## 4. Experiments

### 4.1. Datasets

The VisDrone [24] dataset, which is a small-target dataset of aerial drone scenes, was released in 2018 and further expanded in 2019 by Tianjin University and other institutions. 10,209 images were provided for the small-target detection task, of which 6471 images were used for training, 343,204 were for marked boxes, 548 were for validation, and 3190 were for testing. The image data are characterized by large image resolutions, many small targets, and an average image size of 1002 × 1520.

Figure 5 reveals a notable class imbalance in the dataset, which may bias the model’s generalization capability. Sampling analysis indicates that the dataset encapsulates challenging real-world scenarios prevalent in small-target recognition, including severely occluded objects (e.g., pedestrians obscured by vehicles), low-light targets (e.g., unilluminated nighttime pedestrians and vehicles), and high-density overlapping instances (e.g., crowded urban areas). These conditions closely mirror practical deployment challenges yet remain underexplored in current literature.

The DOTA-v1.5 [25] is a high-quality image data set for ground target detection acquired by satellite. The dataset contains 403,318 target instances in 16 categories. Each image ranges in size from 800 × 800 pixels to 20,000 × 20,000 pixels and contains objects that exhibit a vast variety of scales, orientations, and shapes. It is worth mentioning that this data set is richer in the collection and labeling of small-size objects, and even 10-pixel objects are meticulously labeled. As shown in Figure 6, this is a comparison of the number of targets in different categories in this dataset.

By comparing the different characteristics of the two data sets used in this experiment, we can find that the VisDrone data set is mainly a city image in a complex light environment, in which the objects to be identified are greatly affected by light and seriously blocked. The DOTA-v1.5 dataset is characterized by a large gap between the size of the image and the size of the recognition target, a relatively simple light condition and a complex image background. By summarizing the characteristics of the above two data sets, we can find that all the situations in the target recognition task are included, so the experimental results are of practical significance.

### 4.2. Experimental Details

In the experiments, we used Ubuntu 22.04 as the operating system and Python 3.10.13, PyTorch 2.1.1, and Cuda 11.8 as the software environment. The experiment used an NVIDIA GeForce RTX 3090 24 G graphics card (The graphics card was manufactured by ASUSTeK Computer Inc. (ASUS) (Taipei, Taiwan) and procured within China through the JD.com (Jingdong) e-commerce platform) as hardware. The implementation code of the neural network was modified based on Ultralytics version 8.0.105. The hyperparameters used in the training, testing, and validation of the project were consistent. The training epochs were set to 800, and the image input into the network was rescaled to 640 × 640. All experiments used the same data augmentation operations to more fairly determine the performance of the model, including horizontal and vertical flips and random rotation operations.

### 4.3. Comparison Results

#### 4.3.1. Comparison with State-of-the-Art Methods

Table 1 presents the results of comparative experiments based on the VisDrone dataset and the DOTA-v1.5 dataset.

Through observation, we find that the total parameter count of PGI-ViMamba (27.8 M) is moderately low. This is because our model removes the SPPF and PANET-like structures commonly used in the YOLO series and replaces them with an additional auxiliary detection head and the MLAG PGI structure. In terms of the computational metric FLOPs, the computational cost of our proposed method (209.0 G) is at a moderately high level compared to other models. The increase in computational cost is primarily due to the inclusion of the SS2D module in the PGI-ViMamba network, as its four-directional scanning mechanism contributes to additional computations. Additionally, the use of attention mechanisms in our approach further increases the computational load. In terms of the Frames Per Second (FPS) metric, there is not much difference between PGI-ViMamba and the lightweight YOLOv8s. PGI-ViMamba reaches 153 FPS, a decrease of only 35 FPS, and it is at a moderate level. This is an acceptable result.

By observing the P, R, and mAP_50_ metrics, it can be seen that the results of any method on the DOTA-v1.5 dataset are higher than those on the VisDrone dataset. The reason for this phenomenon lies in the fact that although DOTA-v1.5 has more detection categories and a larger variation in target sizes, its lighting conditions are relatively uniform, and the detected objects are not occluded. Precision (P) measures the proportion of correct predictions made by the model. A higher precision indicates fewer false positives (FP) in the model’s predictions. Specifically, on different datasets, the precision of PGI-ViMamba is 1.1% and 2.1% higher than that of the second-best model, YOLOv11. Recall (R) measures the model’s ability to detect all targets. A higher recall indicates fewer missed detections by the model, corresponding to a reduction in false negatives (FN). On the VisDrone dataset, the recall of PGI-ViMamba is 1.2% lower than that of the best-performing model, YOLOv11x, while on the DOTA-v1.5 dataset, it is 0.4% lower than that of YOLOv11x.

Based on the analysis of the above results, it can be observed that PGI-ViMamba excels at detecting targets with prominent features, which is why its precision is higher than other methods. However, when the target features are less apparent or the noise is excessive, the recall rate of PGI-ViMamba tends to be lower compared to other models. This comprehensive analysis suggests that PGI-ViMamba adopts a cautious approach in target detection, prioritizing the avoidance of false positives over the risk of missing detections. In other words, it tends to be more conservative, preferring to miss some targets rather than incorrectly detect non-targets.

The mAP_50_ is the mean average precision calculated at an IoU (Intersection over Union) threshold of 0.5. It comprehensively evaluates the detection performance of the model across different categories, with an IoU threshold of 0.5 indicating that the overlap between the predicted bounding box and the ground truth box is at least 50%. A higher mAP_50_ signifies better overall detection performance by the model. On both datasets, PGI-ViMamba outperforms the second-best model by 1.0% and 0.5%, respectively.

To provide a more intuitive comparison of model performance, we present Figure 7 based on experimental results from the VisDrone dataset. In the figure, we compare the accuracy, recall, and number of learnable parameters across different models. As observed from the figure, PGI-ViMamba achieves higher accuracy with a moderate number of parameters while maintaining recall rates comparable to other models.

#### 4.3.2. Comparison of Precision and Recall Rate of Different Models

As shown in Figure 8, we collected the training process data of nine models on the VisDrone dataset for comparison, which is conducive to evaluating the model capability of PGI-ViMamba. Using the experimental data, we drew the accuracy and recall curves of different models in 800 training sessions. By observing the change of the curve, we can find that the accuracy and recall rate of different models tended to be stable when the training rounds were between 700 and 800 epochs.

First, analyzing accuracy dynamics, we observe that PGI-ViMamba’s accuracy increases rapidly from epochs 100 to 300, demonstrating a significantly faster improvement rate compared to other models. This indicates PGI-ViMamba’s superior ability to learn the features of small-scale targets early in training. Second, while PGI-ViMamba’s recall rate exhibits a slower ascent than YOLOv10x and YOLOv11x, its final recall performance remains comparable to these models.

From these analyses, PGI-ViMamba emerges as a high-precision, prudent small-target detection algorithm. Given the inherent information decay in extremely small target images, these characteristics align with the demands of small-target detection tasks, enhancing the reliability of deep neural networks in practical applications.

#### 4.3.3. Comparison of Precision and Recall of Different Objects

To further compare the performance of different models, we used experimental results from the more difficult VisDrone dataset to make Table 2 and the radar diagram shown in Figure 9. In the table we list the accuracy and recall rates of different types of targets, as well as the average.

In Figure 9, the left panel displays radar charts illustrating the accuracy rates for ten distinct target recognition categories, while the right panel shows radar charts for their corresponding recall rates.

Observations reveal that PGI-ViMamba outperforms other models in accuracy across all categories, as evidenced by its radar curve fully encompassing those of competing models.

The experimental results of fine-grained feature categories with rich features and high recognition difficulty, such as motorcycles, pedestrians, vans, tricycles, and awning tricycles, demonstrate the significant advantages of using the PGI-ViMamba model. Its performance exceeds that of the second-best model by 0.8%, 1.3%, 1.2%, 2.0%, and 0.8% respectively. For occlusion-prone categories like People and Bicycle, PGI-ViMamba still achieves accuracy improvements of 0.2% and 0.6%. Coarse-grained feature categories with abundant training samples and prominent features (e.g., car, truck, bus) exhibit comparable performance between PGI-ViMamba and other models, with marginal gains of 0.1%, 0.5%, and 0.1% over the second-best model.

In recall comparisons, PGI-ViMamba does not dominate most categories but demonstrates superiority in Car, Van, and Truck—categories with dominant coarse-grained features—outperforming the second-best model by 1.1%, 0.1%, and 0.4%, respectively.

Collectively, these results indicate that PGI-ViMamba prioritizes minimizing false positives over maximizing recall when category features are ambiguous or corrupted, aligning with its design objective of ensuring reliable detection under challenging conditions.

## 5. Ablation Study

In this paper, we conducted two ablation experiments, one of which was to reduce the depth of MLAG PGI to assess the impact on model accuracy. On the other hand, a classical feature extraction network was used to replace our proposed SPD-Conv-VSS module to evaluate the influence of different backbone on the performance of our proposed model.

### 5.1. Comparison of Different Levels of MLAG PGI

The first aspect of the ablation experiment is to reduce the reversible depth of MLAG PGI. The network structure under different reversible depths is shown in Figure 3. From Table 3, we observe that gradually reducing the reversible depth of MLAG PGI will lead to changes in the accuracy and recall rates. It was found that the accuracy and recall rate decreased with the decrease of the reversible depth. It can be seen from the experimental results that the larger MLAG PGI reversible depth is beneficial to the feature extraction of small targets.

The second aspect of the ablation experiment was to verify whether a larger MLAG PGI reversible depth is more conducive to target recognition. When the reversible depth of MLAG PGI was increased to five levels, both the accuracy and recall rate were found to decrease. This shows that a greater reversibility depth of MLAG PGI is not better, so it is necessary to design a suitable MLAG PGI reversibility depth using a neural network model. The reason for this phenomenon is that a too-large MLAG PGI reversible depth may cause the loss of feedback gradient information.

### 5.2. Comparison of Different Information Extraction Backbone Networks

We conducted ablation experiments from two perspectives, with the results summarized in Table 4. First, by maintaining the feature extraction module unchanged, we compared the model’s performance with and without the MLAG PGI module. Significant degradation across all of the evaluation metrics was observed. For instance, when PGI-ViMamba employed the SPD-Conv-VSS feature extraction module without MLAG PGI, Precision (P) decreased by 28.9%, Recall (R) dropped by 17.8%, mAP_50_ declined by 16.4%, and mAP_50:95_ fell by 13.7%. This severe performance deterioration stems from the dual functionality of MLAG PGI: it acts not only as a gradient propagation pathway to stabilize backward signal flow but also as a multi-scale feature fusion channel to enhance hierarchical representation learning, both of which are critical to the model’s effectiveness.

Second, we evaluated the impact of the SPD-Conv-VSS module. When SPD-Conv-VSS was replaced in models equipped with MLAG PGI, the performance suffered at least a 4.7% reduction in P, 10.3% in R, 4% in mAP_50_, and 2.5% in mAP_50:95_. Even without MLAG PGI, replacing SPD-Conv-VSS still led to declines of 3.8% in P, 11.2% in R, 1% in mAP_50_, and 2% in mAP_50:95_. These comparative results unequivocally demonstrate that the SPD-Conv-VSS module significantly contributes to improving detection metrics across diverse experimental configurations.

By systematically analyzing these ablation outcomes, we conclude that both MLA G PGI and SPD-Conv-VSS are indispensable components of the proposed architecture, each addressing distinct challenges in small-target detection while synergistically enhancing overall robustness and precision.

## 6. Conclusions

This study focuses on the information bottleneck problem in deep neural networks. The information bottleneck problem refers to the gradual loss of input information during data propagation through multiple network layers in the training process, which prevents models from fully utilizing effective information from raw data. This phenomenon significantly degrades the performance of the model, especially in small-target recognition tasks, where the impact is more pronounced.

To mitigate information loss caused by the information bottleneck, we designed a novel network architecture termed PGI-ViMamba. The core design philosophy of this model deviates from the conventional “information extraction → information fusion → information spatial mapping” pipeline, instead adopting a parallel multi-branch architecture, which includes two forward information extraction branches and one reverse branch. The two forward branches consist of a primary branch and an auxiliary branch. The primary branch is equipped with a single detection head, while the auxiliary branch incorporates multiple detection heads. Both branches utilize our proposed SPD-Conv-VSS module as the core information extraction component. This module integrates the SS2D module and SPD-Conv through parallel substructures, and its effectiveness has been validated via ablation experiments. Another critical innovation is the introduction of a reverse branch—a specially designed attention-gated pathway called MLAG PGI (Multi-Level Attention-Gated Progressive Gradient Integration). This reversible branch not only generates reliable gradient information but also enables multi-scale feature fusion, ensuring that deep-layer features retain critical image information. To validate MLAG PGI’s effectiveness, two ablation studies were conducted. First, hierarchical impact analysis comparing MLAG PGI at different levels revealed that the 4-level configuration delivers optimal small-target recognition performance. Second, removing MLAG PGI caused significant performance degradation across all tested feature extractors.

The proposed PGI-ViMamba, with 27.8M parameters and 209.0 GFLOPs, achieves 1.1% and 2.1% precision improvements on the VisDrone and DOTA-v1.5 datasets, respectively, while maintaining recall rates comparable to other models. Comparative detection analyses further indicate its reduced tendency for false positives. Our work demonstrates that the Mamba model, through its unique feature extraction mechanism, and the enhanced PGI framework, when synergistically integrated, significantly enhance small-target recognition performance while effectively mitigating the inherent information degradation challenges in deep neural networks.

## Figures and Tables

**Figure 1 sensors-25-02117-f001:**
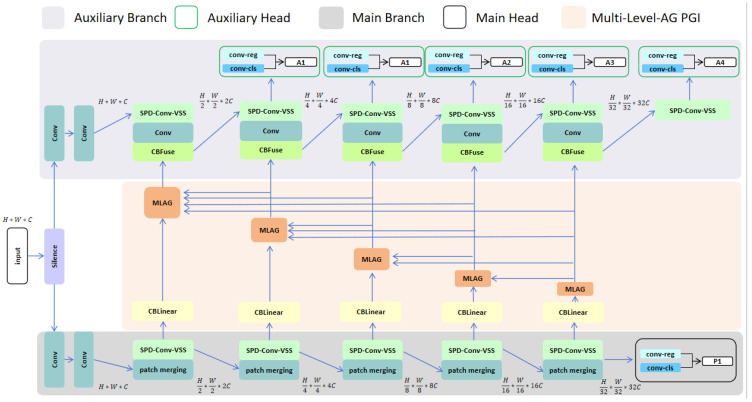
Architecture of PGI-ViMamba.

**Figure 2 sensors-25-02117-f002:**
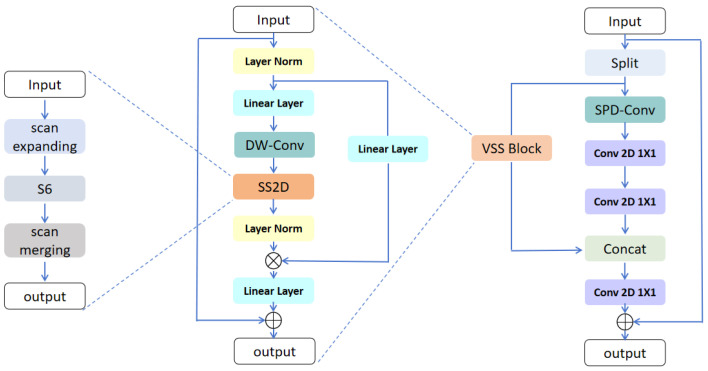
SPD-Conv-VSS module mainly adds branches containing SPD-Conv and SS2D.

**Figure 3 sensors-25-02117-f003:**
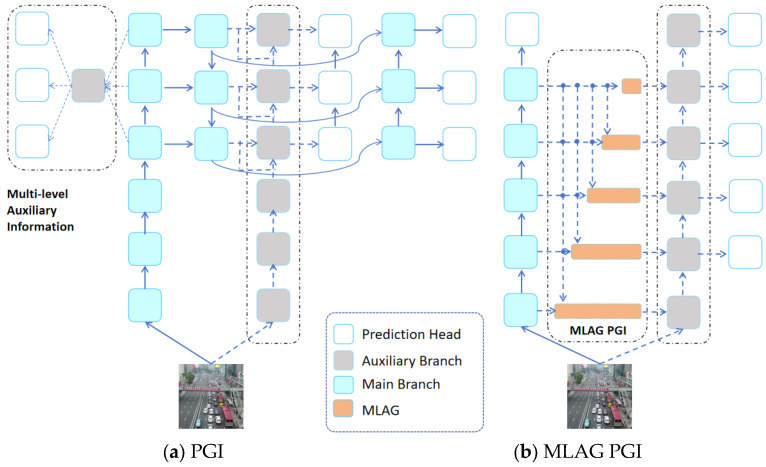
PGI vs. MLAG PGI.

**Figure 4 sensors-25-02117-f004:**
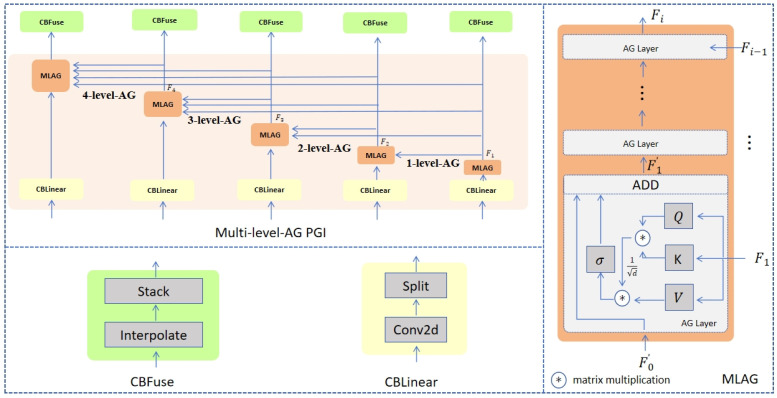
Multi-Level-Attention-Gated Programmable Gradient Information (MLAG PGI).

**Figure 5 sensors-25-02117-f005:**
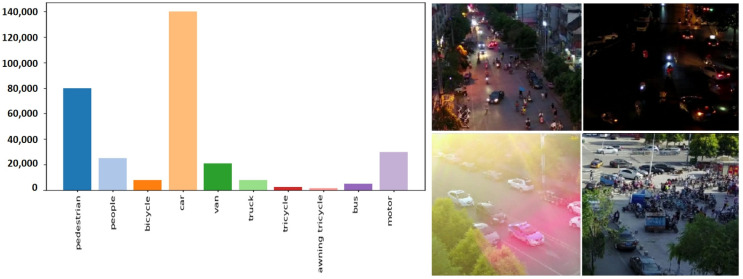
VisDrone Dataset.

**Figure 6 sensors-25-02117-f006:**
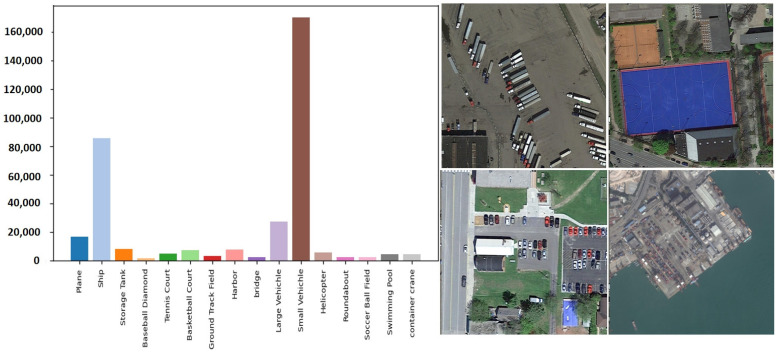
DATA-v1.5 Dataset.

**Figure 7 sensors-25-02117-f007:**
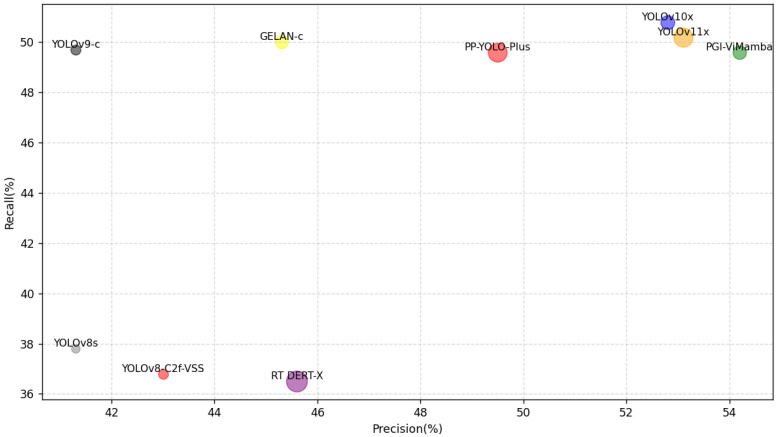
Comparison of the precision, recall, and parameter numbers of different models.

**Figure 8 sensors-25-02117-f008:**
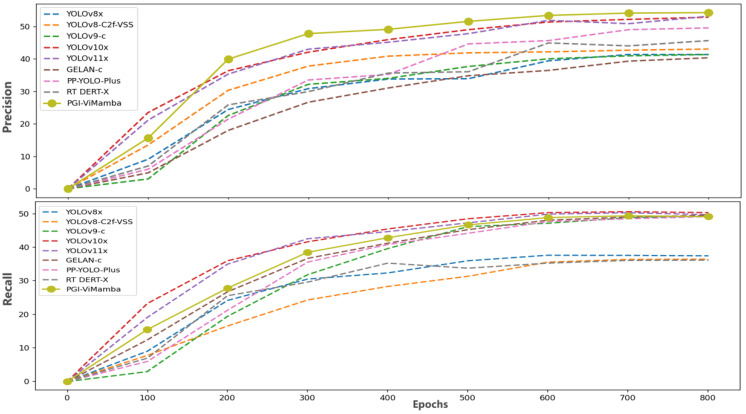
Comparison of precision and recall rate of different models.

**Figure 9 sensors-25-02117-f009:**
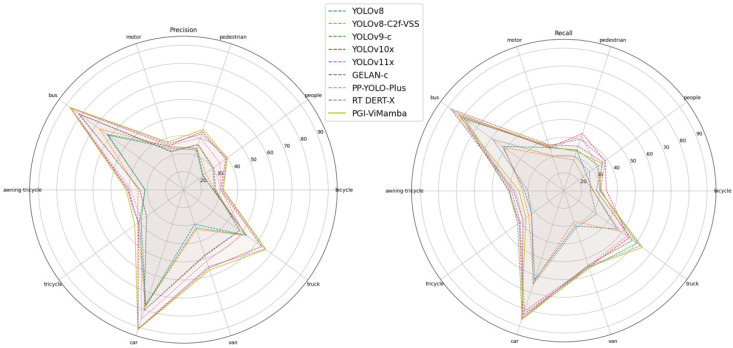
Comparison of precision and recall of different object.

**Table 1 sensors-25-02117-t001:** Result of comparison experiments on the VisDrone dataset and DOTA-v1.5 dataset.

Model	Param. (M)	FLOPs (G)	P (%)	R (%)	mAP_50_ (%)	FPS
YOLOv8s	11.2	28.6	41.3/60.1 *	37.8/58.3	22.1/25.6	**188**
YOLOv8-C2f-VSS	14.9	35.9	43.0/63.8	36.8/57.8	35.9/38.9	164
YOLOv9-c	25.3	102.1	41.3/61.3	49.7/63.4	41.3/42.5	150
YOLOv10x	29.5	160.4	52.8/70.1	**50.8**/**66.7**	40.8/43.6	84
YOLOv11x	56.9	194.9	53.1/72.0	50.2/65.4	41.6/44.8	92
GELAN-c	25.5	102.8	45.3/63.2	50.0/66.2	38.3/41.6	160
PP-YOLO-Plus	54.6	115.8	49.5/68.2	49.6/53.3	42.1/43.8	150
RT DERT-X	67.0	234.1	45.6/61.8	36.5/50.3	42.5/43.6	148
PGI-ViMamba	27.8	209.0	**54.2/74.1**	49.6/66.3	**43.5/45.3**	153

* Result on the VisDrone dataset/Result on the DOTA-v1.5 dataset. In the table, the bold indicates the experimental results of the best-performing model, while the underline denotes the results of the second-best model.

**Table 2 sensors-25-02117-t002:** Comparison results of different categories of VisDrone datasets in different models.

Category	YOLOv8	YOLOv8-C2f-VSS	YOLOv9-c	YOLOv10x	YOLOv11x	GELAN-c	PP-YOLO-Plus	RT DERT-X	PGI-ViMamba
Bicycle	26.3/26.4	27.3/26.3	26.6/30.9	30.6/**33.8**	31.5/30.5	27.2/28.2	30.6/31.3	27.9/24.9	**32.1**/30.1
People	23.5/25.5	25.2/23.2	23.1/33.9	38.9/**38.5**	39.7/38.7	30.1/37.1	35.1/32.9	30.8/27.8	**39.9**/36.2
Pedestrian	33.9/33.9	34.1/28.1	33.1/33.9	43.8/**43.7**	42.4/40.4	36.2/36.2	40.2/42.3	36.3/30.3	**45.1**/33.1
Motor	35.2/35.4	35.2/30.2	34.2/35.5	35.7/35.6	36.9/36.6	32.1/**37.1**	35.7/34.7	32.5/30.5	**37.7**/36.2
Bus	62.3/52.3	67.3/51.3	61.3/79.3	86.8/81.2	87.6/77.6	81.1/**88.1**	86.9/86.1	81.9/59.2	**87.7**/83.1
Awning-tricycle	31.1/31.9	35.1/33.1	31.3/40.3	40.9/**40.8**	40.2/39.2	34.1/38.1	39.1/37.1	34.1/30.1	**41.7**/35.1
Tricycle	40.2/32.2	41.2/34.2	38.8/**41.8**	40.8/40.7	41.1/40.1	35.2/38.2	38.8/38.1	35.2/32.2	**43.2**/36.2
Car	77.4/62.7	78.2/64.4	77.4/85.4	91.4/83.4	90.9/81.3	80.4/85.4	85.6/85.4	80.4/65.4	**91.5**/**86.5**
Van	30.1/30.9	33.2/28.2	32.2/55.9	55.6/55.1	54.8/54.8	48.1/54.1	51.6/54.6	48.6/32.1	**56.8**/**55.2**
Truck	53.1/46.7	53.1/49.1	52.9/60.4	63.5/55.1	65.9/62.9	48.5/57.5	51.5/53.5	48.5/32.5	**66.4**/**64.3**
Avg *	41.3/37.8	43.0/36.8	41.3/49.7	52.8/**50.8**	53.1/50.2	45.3/50.0	49.5/49.6	45.6/36.5	**54.2**/49.6

Avg * represents the average of all categories of the identified target. The numbers in front of “/” are Precision, and the numbers after “/” represent Recall rates. In the table, the bold indicates the experimental results of the best-performing model, while the underline denotes the results of the second-best model.

**Table 3 sensors-25-02117-t003:** The results of the comparison of different levels of MLAG PGI.

n-Level-PGI	P (%)	R (%)	mAP_50_ (%)	mAP_50:95_ (%)
0-level-PGI ^1^	20.3	31.8	20.1	14.1
1-level-PGI	21.9	37.8	28.9	23.2
2-level-PGI	29.3	40.9	35.3	25.5
3-level-PGI	45.3	41.7	38.3	28.6
4-level-PGI	**54.2**	**49.6**	**43.5**	**32.3**
5-level-PGI	41.5	41.8	34.1	20.3

^1^ 0-level-PGI indicates that programmable gradient information is not added. In the table, the bold indicates the experimental results of the best-performing model.

**Table 4 sensors-25-02117-t004:** The results of the comparison of different BackBones.

Model	BackBone	MLAG ^1^	P (%)	R (%)	mAP_50_ (%)	mAP_50:95_ (%)
PGI-ViMamba	EfficientNet	**X**	20.3	10.8	10.1	10.1
PGI-ViMamba	EfficientNet	**√**	38.5	39.3	31.9	19.2
PGI-ViMamba	ResNet-50	**X**	20.3	17.9	19.3	5.5
PGI-ViMamba	ResNet-50	**√**	35.3	38.1	33.3	18.6
PGI-ViMamba	VGG-16	**X**	21.5	20.6	26.1	17.3
PGI-ViMamba	VGG-16	**√**	49.5	33.9	39.5	30.5
PGI-ViMamba	SPD-Conv-VSS	**X**	**25.3**	**31.8**	**27.1**	**19.3**
PGI-ViMamba	SPD-Conv-VSS	**√**	**54.2**	**49.6**	**43.5**	**33.0**

^1^ MLAG specifies whether to use MLAG PGI. MLAG PGI is 4-Level-PGI by default. In the table, the bold indicates the experimental results of the best-performing model, while the underline denotes the results of the second-best model. In the table, “X” indicates that MLAG is not used in the experiment, and “√” indicates that MLAG is used in the experiment.

## Data Availability

The data presented in this study are available on request from the corresponding author. The data are not publicly available due to privacy restrictions.

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
