# Peer review of "An Innovative Small-Target Detection Approach Against Information Attenuation: Fusing Enhanced Programmable Gradient Information and a Novel Mamba Module"

_sensors, 2025, doi:10.3390/s25072117_

Round 1
Reviewer 1 Report
Comments and Suggestions for Authors
The authors introduced the PGI-ViMamba model, which successfully tackles the challenge of information bottlenecks in deep learning models, significantly enhancing the accuracy and efficiency of small object detection. Nevertheless, there are still several aspects of the paper that could be improved:
1. Please further verify the numbering of the figures in the paper. For example, should Figure 1 in line 238 be corrected to Figure 4?
2. Regarding the 200 training epochs mentioned in section 4.2, could the authors explain why only 200 epochs were used to train the model, instead of more? For such a large-scale dataset, I believe 200 epochs may not be sufficient to achieve the model's optimal performance. As shown in Figure 6, for most models, the precision and recall are still increasing after 100 epochs and have not stabilized.
3. Is the statement "Tables may have a footer" in line 256 accidentally included, or is it necessary? Please verify.
4. There is confusion in the numbering of the formulas in the paper. For example, there are two different formulas labeled as formula 10. Please carefully check and correct all formula numbers in the paper.
5. Section 4.2 mentions that the model was trained for 200 epochs, but the figures and descriptions in section 4.3.2 use 100 epochs. Please verify this data inconsistency. Accuracy in data is essential for scientific research.
6. Regarding Figure 7, I suggest including a corresponding table to help readers better understand and interpret the information.
Author Response
Comments 1: Please further verify the numbering of the figures in the paper. For example, should Figure 1 in line 238 be corrected to Figure 4?
Response 1: Agree. I have modified the picture number you proposed. The numbers of the graphs and tables in the full text are checked.
Comments 2: Regarding the 200 training epochs mentioned in section 4.2, could the authors explain why only 200 epochs were used to train the model, instead of more? For such a large-scale dataset, I believe 200 epochs may not be sufficient to achieve the model's optimal performance. As shown in Figure 6, for most models, the precision and recall are still increasing after 100 epochs and have not stabilized.
Response 2: Agree.
Thank you for your insightful feedback. We agree that extending training epochs is critical for model convergence. Following your suggestion, we retrained all models for 800 epochs and observed that metrics (precision, recall) stabilized after an initial rise, confirming your observation. The updated results demonstrate improved performance, particularly for small and occluded targets. We have revised the manuscript to reflect these changes, including extended training protocols and stabilized metric curves. Your comment has significantly strengthened the empirical validity of our work.
The changes made in the manuscript are on page 10, table 1 and page 14,table3 and page 14 ,table 4.
Comments 3: Is the statement "Tables may have a footer" in line 256 accidentally included, or is it necessary? Please verify.]
Response 3: Agree.
The footer of the table has been modified to introduce the meaning of the data.
The changes made in the manuscript are on page 10, lines 320.
Comments 4: There is confusion in the numbering of the formulas in the paper. For example, there are two different formulas labeled as formula 10. Please carefully check and correct all formula numbers in the paper.]
Response 4: Agree.
I have modified the situation of formula 10 marking confusion, and proofread and adjusted the formula marking in the whole paper.
Comments 5: Section 4.2 mentions that the model was trained for 200 epochs, but the figures and descriptions in section 4.3.2 use 100 epochs. Please verify this data inconsistency. Accuracy in data is essential for scientific research.
Response 5: Agree.
We have reconducted the experiment in the paper, increased the training rounds to 800 epochs, and updated the figure in section 4.3.2 to ensure that the data are consistent.
The changes made in the manuscript are on page 12, Figure 8.
Comments 6: Regarding Figure 7, I suggest including a corresponding table to help readers better understand and interpret the information.
Response 6: Agree.
We have redrawn the graph problem you pointed out and added the corresponding data table. At the same time, nine models can be compared and analyzed.
The changes made in the manuscript are on page 13, Table 2 and Figure 9.
Reviewer 2 Report
Comments and Suggestions for Authors
- The abbreviation is not appropriate in the title.
- In Abstract, all abbreviations should be given their full name where they first appear.
- As declared in line 121, actually, YOLO v9 is not the latest generation. The statement should be modified. YOLOv11 proposed in 2024 has introduced C2PSA and C3k2 to increase the ability to detect small targets or masked targets. YOLOv10 has also introduced some innovations to achieve the same goal. So it is necessary for authors to add a series of comprehensive comparative studies with YOLOv10, YOLOv11 and their variants.
- In this paper, the GELAN is not involved in the proposed method, and innovations, so it is suggested to introduce PGI only in line 125, and the statement of GELAN is unnecessary.
- In line 141, it is suggested to replace SS2D by SPD-Conv-VSS.
- In line 148, is the backbone network still in the proposed network? As stated in 140, the proposed network does not contain the bottleneck structure.
- As shown in paper, in line 150, the main trunk network and the main branch may be the same module. It is necessary to make the statement of the same thing consistent throughout this paper.
- The writing of equations 1 and 2 should be modified.
- The explanation of equations 3 to 10 is not enough, please rewrite this part to make the function and procedure of S6 module clearer.
- The full name of S6 should be placed at the location that the S6 first appears, but not in line 206.
- In this paper, the expression of "reversible branch" was used sometimes, while the expression of "invertible branch" was also used. Please make the expression of the same thing consistent throughout this paper.
- For Figure 3, it is suggested to draw the comparison of traditional PGI and the improved PGI, emphasizing on the difference, to make the contribution or innovation clearer and more intuitionistic.
- The equations 10 to 13 lack introductions. The explanation of these equations should be refined. How many learnable matrixes in these equations? Three or Four? And it is necessary to show details of these matrixes, such as the definition, value, and corresponding reasons.
- Equations 10 to 13 cannot be verified by Figure 3. Please reorganize and rewrite this part.
- In Experiments section, the Figure 1 has nothing to do with the balance of object types. Whether this is a clerical error, it is recommended that the author carefully check the figure in the full text to avoid such errors.
- In section 4.3.1, for Table 1, the description is inconsistent with the data in Table 1, it is recommended that the author carefully check and review.
- In section 4.3.1, the analysis of the conclusion is too thin to support the corresponding conclusion. It is suggested that the author enrich the analysis.
- In section 4.3.2, why only four models are selected for comparison here, and why not other algorithms in Figure 5?
- In section 4.3.4, the analysis is too weak, and some key information points in Figure 7 have not been analyzed in detail. It is suggested that the author enrich the analysis in section 4.3.4 in order to better verify the superiority of the proposed algorithm. And, why were three models selected for comparison in this part?
- The conclusion section is somewhat brief and does not effectively highlight the core contribution of the article.
The writing quality of this paper should be improved.
Author Response
Comments 1: [The abbreviation is not appropriate in the title.]
Response 1: Agree.
We have changed the abbreviation in the title to the full name.
The changes made in the manuscript are on page 1, lines 2 through 4.
Comments 2: [In Abstract, all abbreviations should be given their full name where they first appear.]
Response 2: Agree.
We have changed the abbreviations in the abstract to the full name, and changed the same kind of questions.
The changes made in the manuscript are on page 1, lines 18 .
Comments 3: [As declared in line 121, actually, YOLO v9 is not the latest generation. The statement should be modified. YOLOv11 proposed in 2024 has introduced C2PSA and C3k2 to increase the ability to detect small targets or masked targets. YOLOv10 has also introduced some innovations to achieve the same goal. So it is necessary for authors to add a series of comprehensive comparative studies with YOLOv10, YOLOv11 and their variants.]
Response 2: Agree.
We have modified the sentences in this paragraph and added a comparison with YOLOv10 and YOLOv11 in the experiment
The changes made in the manuscript are on page 3, lines 130 through 141.
Comments 4: [In this paper, the GELAN is not involved in the proposed method, and innovations, so it is suggested to introduce PGI only in line 125, and the statement of GELAN is unnecessary.]
Response 4: Agree.
We have removed the description of the GELU and rewritten the relevant parts.
The changes made in the manuscript are on page 3, lines 130 through 141.
Comments 5: [In line 141, it is suggested to replace SS2D by SPD-Conv-VSS.]
Response 5: Agree.
Thank you for your constructive suggestion. We fully agree with your observation and have revised line 141 by replacing SS2D with SPD-Conv-VSS, as recommended.
The changes made in the manuscript are on page 4, lines 151.
Comments 6: [In line 148, is the backbone network still in the proposed network? As stated in 140, the proposed network does not contain the bottleneck structure.]
Response 6: Agree.
Thank you for pointing out this discrepancy. The term "backbone network" in line 148 is indeed a clerical error and has been corrected to "main branch" in order to be consistent with our architectural description. To clarify, our approach does not use the traditional "compression-expand" bottleneck structure. Instead, we propose a two-branch compression framework (main branch + auxiliary branch) with an integrated reversible attention gate (MLAG-PGI). The design avoids the dimensionality reduction of the critical path and ensures the fine-grained feature retention and bidirectional gradient flow of the small target. We appreciate your careful review, which improves the accuracy of our method descriptions.
The changes made in the manuscript are on page 4, lines 158.
Comments 7: [As shown in paper, in line 150, the main trunk network and the main branch may be the same module. It is necessary to make the statement of the same thing consistent throughout this paper.]
Response 7: Agree.
We have revised this issue to maintain the consistency of the statement throughout the paper. The changes made in the manuscript are on page 4, lines 149 through 161.
Comments 8: [The writing of equations 1 and 2 should be modified.]
Response 8: Agree.
The original formulas 1 and 2 had errors and we have made changes to them.
The changes made in the manuscript are on page 5, lines 169 through 184.
Comments 9: [The explanation of equations 3 to 10 is not enough, please rewrite this part to make the function and procedure of S6 module clearer.]
Response 9: Agree.
We have redescribed the principle of the S6 module, which is more clear and logical than the original writing.
The changes made in the manuscript are on page 6, lines 205 through 230,equations 3 to 12.
Comments 10: [The full name of S6 should be placed at the location that the S6 first appears, but not in line 206.]
Response 10: Agree.
We've used the full name where the S6 first appeared, and abbreviations in other locations.
The changes made in the manuscript are on page 5, lines 186 through 187.
Comments 11: [In this paper, the expression of "reversible branch" was used sometimes, while the expression of "invertible branch" was also used. Please make the expression of the same thing consistent throughout this paper.]
Response 11: Agree.
We already use "reversible branch" uniformly, not "invertible branch" throughout this paper.
Comments 12: [For Figure 3, it is suggested to draw the comparison of traditional PGI and the improved PGI, emphasizing on the difference, to make the contribution or innovation clearer and more intuitionistic.]
Response 12: Agree.
We added a comparison between traditional PGI and the improved PGI, detailing the differences between the two.
The changes made in the manuscript are on page 7, Figure 3 and lines 241 through 256.
Comments 13: [The equations 10 to 13 lack introductions. The explanation of these equations should be refined. How many learnable matrixes in these equations? Three or Four? And it is necessary to show details of these matrixes, such as the definition, value, and corresponding reasons.]
Response 13: Agree.
We have modified the corresponding formula and should give a detailed explanation of the formula. There were problems with the previous writing, and I have modified it.
The changes made in the manuscript are on page 8, formula 13 through 17,lines 267 through 273.
Comments 14: [Equations 10 to 13 cannot be verified by Figure 3. Please reorganize and rewrite this part.]
Response 14: Agree.
There were some errors in the figure and formula before, and I have modified them.
The changes made in the manuscript are on page 8, Figure 4,formula 13 through 17.
Comments 15: [In Experiments section, the Figure 1 has nothing to do with the balance of object types. Whether this is a clerical error, it is recommended that the author carefully check the figure in the full text to avoid such errors.]
Response 15: Agree.
Thank you for pointing out the problem, this question is my clerical error. I've already rewritten this paragraph and checked the figure in the full text to avoid such errors
The changes made in the manuscript are on page 9, ,lines 284 .
Comments 16: [In section 4.3.1, for Table 1, the description is inconsistent with the data in Table 1, it is recommended that the author carefully check and review.]
Response 16: Agree.
We have made changes to section 4.3.1 to correct the existing data problems.
The changes made in the manuscript are on page 8, lines 319 through 367.
Comments 17: [In section 4.3.1, the analysis of the conclusion is too thin to support the corresponding conclusion. It is suggested that the author enrich the analysis.]
Response 17:Agree.
We have revised the data analysis in section 4.3.1 to make the analysis more in-depth and detailed.
The changes made in the manuscript are on page 10, lines 319 through 367.
Comments 18: [In section 4.3.2, why only four models are selected for comparison here, and why not other algorithms in Figure 5?]
Response 18: Agree.
I have rewritten this chapter, adding experiments and redrawing all previous comparison models. The experimental results are analyzed in detail in the description.
The changes made in the manuscript are on page 12, lines 368 through 386.
Comments 19: [In section 4.3.4, the analysis is too weak, and some key information points in Figure 7 have not been analyzed in detail. It is suggested that the author enrich the analysis in section 4.3.4 in order to better verify the superiority of the proposed algorithm. And, why were three models selected for comparison in this part?]
Response 19: Agree.
I have rewritten this chapter, adding experiments and redrawing all previous comparison models. The experimental results are analyzed in detail in the description.
The changes made in the manuscript are on page 12, lines 387 through 363.
Comments 20: [The conclusion section is somewhat brief and does not effectively highlight the core contribution of the article.]
Response 20: Agree.
We have revised the Conclusion section to systematically cover the addressed problem, the main design philosophy of the method, the two innovative structures, and the validation of each module's effectiveness.
The changes made in the manuscript are on page 15, lines 464 through 497.
Reviewer 3 Report
Comments and Suggestions for Authors
The author proposes a PGI-ViMamba model, which combines Multi-level Attention Gated PGI (MLAG PGI) and the SPD-Conv-VSS module. This integration approach is innovative in the field of small target detection.
The manuscript is relevant to the field and presented in a well-structured manner. It is a topic of interest to the researchers in the related areas but the paper needs some improvement before acceptance for publication. My detailed comments are as follows:
- In lines 44-62 of the introduction section, it is recommended to add relevant references for the three methods mentioned to overcome the challenges of small target detection.
- The experiments have so far only been conducted on the VisDrone dataset. Although this dataset is somewhat representative, it is suggested that the generalization ability and adaptability of the model be more comprehensively evaluated by conducting experimental validation on other small target detection datasets of different types, such as the small target portion of the COCO dataset and other scene datasets captured by drones.
- In the experimental section, there is a lack of comparison of detection speed metrics such as FPS between PGI-ViMamba and other networks.
Minor editing of English language required.
Author Response
Comments 1: [In lines 44-62 of the introduction section, it is recommended to add relevant references for the three methods mentioned to overcome the challenges of small target detection.]
Response 1: Agree.
I have partially rewritten this chapter and added references.
The changes made in the manuscript are on page 2, lines 45 through 70.
Comments 2: [The experiments have so far only been conducted on the VisDrone dataset. Although this dataset is somewhat representative, it is suggested that the generalization ability and adaptability of the model be more comprehensively evaluated by conducting experimental validation on other small target detection datasets of different types, such as the small target portion of the COCO dataset and other scene datasets captured by drones.]
Response 2: Agree.
We added experiments in the modification process, used another data set DOTA-v1.5, and wrote the experimental results into Table 1.
The changes made in the manuscript are on page 10, Table 1 and page 10 line 319 through 367.
Comments 3: [In the experimental section, there is a lack of comparison of detection speed metrics such as FPS between PGI-ViMamba and other networks.]
Response 3: Agree.
Experiments were added during the modification process, and the experimental results of FPS were written into Table 1.
The changes made in the manuscript are on page 10, Table 1,line 331 through 334.
Round 2
Reviewer 1 Report
Comments and Suggestions for Authors
Accept in present form
Reviewer 2 Report
Comments and Suggestions for Authors
None.
Comments on the Quality of English LanguageSome expressions should be improved.
Reviewer 3 Report
Comments and Suggestions for Authors
NO
Comments on the Quality of English LanguageMinor editing of English language required.